# Utilisation of Mass and Night Ventilation in Decreasing Cooling Load Demand

**Michael Darmanis [1,2,\*], Murat Çakan [1], Konstantinos P. Moustris [2] , Kosmas A. Kavadias [3] and Konstantinos-Stefanos P. Nikas [2,4]**

[1]   Faculty of Mechanical Engineering, Istanbul Technical University, Beyoğlu, Istanbul 34437, Turkey;
      cakanmu@itu.edu.tr

[2]   Laboratory of Fluid Mechanics, Department of Mechanical Engineering, University of West Attica,
      GR12244 Athens, Greece; kmoustris@uniwa.gr (K.P.M.); ksnikas@uniwa.gr (K.-S.P.N.)

[3]   Laboratory of Soft Energy Applications & Environmental Protection, Department of Mechanical
      Engineering, University of West Attica, GR12244 Athens, Greece; kkav@uniwa.gr

[4]   Laboratory of Heat Transfer, Department of Mechanical Engineering, University of West Attica,
      GR12244 Athens, Greece

\*   Correspondence: michaledarm@gmail.com; Tel.: +30-697-809-1358

**Abstract:** The building sector consumes 36% of the world's energy and produces around 40% of energy-related carbon emissions. While the building industry moves towards a zero net greenhouse-gas emission policy, ventilation is, and will be, a necessity for the preservation of air quality—especially in climates defined by unsavoury conditions. Therefore, a "mixing mode" cooling system was employed to lower the required energy consumption at an earthen building situated in the premises of Istanbul Technical University. A room of the high-mass earthen building was monitored under different ventilation and shading conditions. Night ventilation was conducted using two modes, 3.2 and 2.3 air changes per hour, and the air conditioning unit, operating from 08:00 to 17:00, had a set temperature of 23 °C. Night ventilation was somewhat impactful, reducing the average expected cooling energy demand up to 27%. Furthermore, the earthen building proved to be extremely effective on moderating extremes of temperature under non-ventilated conditions. During a rather hot day, with an outdoor maximum temperature of 35 °C, the indoor maximum temperature of the high-mass building was only 25 °C, namely within thermal comfort levels. The diurnal temperature proved to be key in the effective application of night ventilation.

**Keywords:** cooling demand; cooling systems; earthen materials; mixed-mode cooling systems; night ventilation; passive cooling-techniques; thermal comfort

## 1. Introduction

Passive cooling systems have been, and still are, the subject of intensive investigation. Their main interest lies in the significant improvement of thermal comfort they can induce while minimising energy costs. The term "passive" is used in the same sense as in Givoni's exposition of passive cooling strategies [1]; that is, the emphasis in utilising natural cooling resources for the rejection of heat from a building, and in the special case of collaboration of cooling systems, that the heat transfer system is efficient and low-cost.

One (of the several) passive and low-energy cooling system(s) is the cooling of the structural mass by ventilation during the night, namely, nocturnal convective cooling; other means of passive cooling methods include (i) comfort ventilation, (ii) nocturnal radiant cooling, (iii) indirect evaporative cooling, and (iv) utilization of the soil as a cooling source [1,2].

In this work, nocturnal convective cooling will be the subject of main interest, and shall henceforth be referred to as night ventilation. Buildings ventilated at night experience a continuous cooling, by convection, of their structural mass. The following day, this cooled mass may serve as a heat tank to absorb, by natural convection and radiation, the heat penetrating into and generated inside these buildings [1]. When coupled with an air-conditioning unit, this heat dissipation may (and often can) result in energy conservation and reduce summer peak demand for electricity.

Such couplings, also known as "mixed mode" cooling systems [2], have also been suggested by Givoni [1]. Collaborations of this kind are particularly attractive for the special case of combining night ventilation and mechanical cooling, as the latter requires high airflow rates. Thus, night ventilation does not further burden the construction's initial cost, as stated by Blondeau et al. [3].

Night ventilation can be applied to all types of buildings, provided that they are well-insulated and their interior mass enables sufficient cooling potential. Low-mass buildings, however, even when ventilated at night, tend not to retain enough cool reserve to significantly reduce the rate of temperature rise during the daytime, and may actually act as heat traps [4,5].

Efficiency of night ventilation, in decreasing cooling energy demand and making for improved indoor conditions for occupants, rests upon three main considerations [1,6–8]: the climatic parameters, such as diurnal temperature range, relative humidity, and topography; building parameters, such as useful thermal inertia and functionality; and last but not least, technical parameters, such as optimal operation, operation time, air flow rates, and control techniques.

Adopting passive and low-energy cooling systems has become a pressing subject. This is a direct consequence of the desire to conserve energy and to reduce summer peak demand for electricity caused by air-conditioning. However, there is also the pollution argument to consider. In 2015, at a meeting in Paris, world leaders pledged to move towards a zero net greenhouse gas emission policy. This is a challenging task, and the building industry makes it even more so. Buildings consume 36% of the world's energy and produce around 40% of energy-related carbon emissions. Cement manufacturing contributes 6% to the world's carbon emissions. Steel, a large quantity of which goes into buildings, accounts for another 8% [9].

More recycled steel and reduction of lorry journeys, by prefabrication in off-site factories, may make buildings "greener". However, no other building material has environmental credentials as exciting and overlooked as adobe. Adobes are dried mud or unburnt bricks that have been used in the construction of dwellings and other structures for thousands of years, and bear a near-zero carbon footprint [10]. The term is usually used to refer to various building materials whose base is either mud or soil [11].

The defining property of adobe (or earthen) buildings lies in their unique ability to store cooling loads. They are known to reduce cooling demand during hot summers due to the thermal inertial effect that they may induce [12]. The outdoor temperature both rises above and falls below the indoor temperature. The same, however, is not true for the winter season. During cold winters, the indoor temperature always remains above the outdoor temperature, therefore increasing heat demand.

The present study deals with the collaboration of mass and night ventilation as a way to improve comfort or reduce the cooling loads of an earthen building in summer. It first presents the main results of the full-fledged experiment, the aim of which was to check the effectiveness of night ventilation by noting the decrease of cooling energy. The impact of the environment's diurnal temperature and the air flow rate of night ventilation are discussed. Then, the structure's natural behaviour is assessed by monitoring it without the use of any cooling applications.

Before moving forward, some of the published contributions that bear directly on the effectiveness of similar cooling systems need to be discussed. Numerous studies, both numeric and experimental in nature, have demonstrated the ability of night ventilation to improve comfort conditions [5,7,13–15]. Shaviv et al. [5] argued that significant comfort conditions could be achieved with up to 20 air changes per hour, while Agas et al. [13] and Blondeau et al. [7] could not indicate any significant comfort improvement above 8 ach to 10 ach.

Blondeau et al. [3] experimented in a high-mass building, whose fabric was made of concrete, and showed that night ventilation decreased as the required indoor temperature increased. For night ventilation intensity of 8 ach, the calculated mechanical energy reduction varied from 12% to 56% for an air-conditioning set point of 22 °C to 26 °C; whereas Geros et al. [14], working with a light-mass structure made of concrete, showed that for a 28 °C set point, the calculated conservation varied from 14% to 36% for 5 ach to 30 ach. It was also estimated that 30 ach could reduce the cooling load demand by up to 71%. A large-scale experiment on residential buildings (covering a vast spectrum of cooling needs and applied flow rates) was conducted by Santamouris et al. [4]. Energy savings, with an average of 12 kW h m$^{-2}$ y$^{-1}$, could be achieved with the use of night cooling. On a different note, Shaviv et al. [5] and Givoni [16] showed that night ventilation had only a very small effect on the indoor maxima of low-mass buildings. Particularly, the former pointed out that light structures had the tendency to behave like heat traps; in many instances, the indoor maxima was higher than the outdoor one.

## 2. Materials and Methods

### 2.1. Full-Fledged Experiment

The earthen building (see Figure 1a on page 3), situated at ITU Ayazaga Kampusu, Istanbul (41°6′2″ N 29°1′16″ E) is made of alker, a mud-based building material [17], often designated as stabilized adobe. It bears a density ($\rho$) of 1620 kg m$^{-3}$ and a specific heat capacity ($C_p$) of 1.2 kJ kg$^{-1}$ K$^{-1}$. Conductivity measurements conducted by Kafescioğlu [18] yielded an R-value of approximately 2 m$^2$ K W$^{-1}$ and a thermal conductivity ($\lambda$) of 0.5 W m$^{-1}$ K$^{-1}$. Alker is an improved version of adobe. It's significantly tougher, more resilient to water, and less susceptible to cracking. Stabilized adobe bricks contain enough stabilizer (in this case, 2.5% to 5% of slaked lime) so as to limit water absorption of the brick.

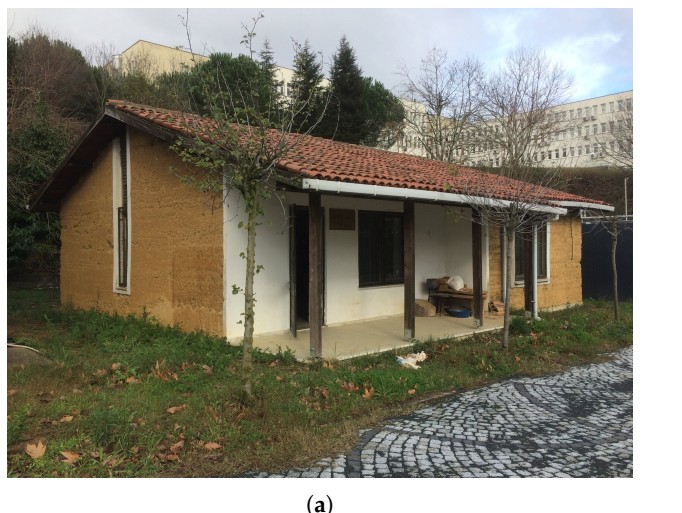
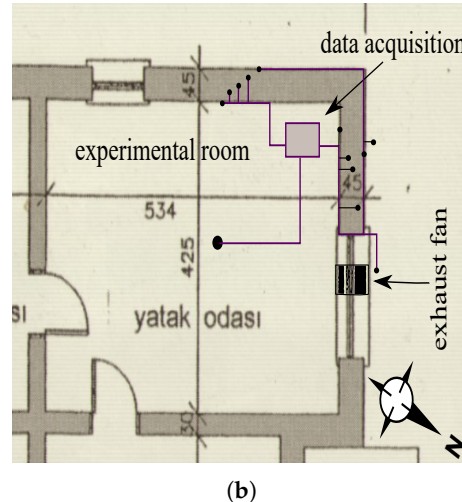

(**a**)　　　　　　　　　　　　　　　　　　　　　　　　(**b**)

**Figure 1.** Earthen building. (**a**) Side view, (**b**) room plan [18].

The interior walls of the building had a width of 30 cm and the outer walls were 45 cm thick, while their inner surface was covered with normal gypsum plaster. The night ventilated zone composed of a room with two windows (one in each wall) and a volume of 44.93 m$^3$. The doorways of the two walls, adjacent to the room under investigation, were blocked using a plastic cloth and house furniture. The directions of the walls in contact with the environment were 154° south-east and 59° north-east, respectively.

To perform night ventilation, a fan was installed in the northeastern window (see Figure 1b on page 3) of the building, with the windows of the experimental room being

shut. Using a set point of 23 °C, the air conditioning unit (of approximately 7000 BTU) operated from 08:00 until 17:00 on a daily basis.

A total of 13 thermocouples were used for the measurements of (i) the air temperature of the room and of the environment, (ii) the operating temperature of the fan during night ventilation, and (iii) five measurements across the building's blocks for each of the two external walls, respectively. All the thermocouples were made from the same metallurgical batch of K-type (Chromel-Alumel) thermocouple wire, and were TIG welded in-house. A computer-based data acquisition system (Keithley 2700) was employed to acquire the thermocouples' output. Adequate adhesion of the sensors with the walls' surfaces was ensured through the use of duck tape, thus preventing thermal bridging. For the indoor temperature, a thermocouple was welded upon a metal mass (approximately 50 g) located at the center of the room.

The time interval between two measurements was 15 min, and the average absolute error according to the manufacturer was 0.5 °C. However, a reliability test showed that all the thermocouples were giving the same temperature within an uncertainty of 0.4 °C. A power meter was used to measure the electricity consumption of the air-conditioning unit.

In its normal configuration, the maximum air flow given by the fan manufacturer was $140 \, \mathrm{m^3 \, h^{-1}}$, roughly equivalent to 3 ach. This value was then verified by airspeed measurements using an anemometer (Testo Term type 4500/4510). Measurements of the air flow rate with each fan speed were performed. The average air speed in front of the fan (from the inside) was measured at six points, twice. The two measurements of each point were then combined as weighted averages (see Taylor [19]). Thus, each outlet speed is the average of six measurements at each speed. The air flow rate and the air changes per hour were calculated by taking into account the area of the fan's opening. At medium speed ($4.7 \, \mathrm{m \, s^{-1}}$), the fan yielded 2.3 air changes per hour, while at full speed ($6.7 \, \mathrm{m \, s^{-1}}$), there were 3.2 air changes per hour.

Several schemes were monitored in order to assess the thermal behaviour of the building. The following operating conditions were employed at a sequence of experiments, spanning a total of 24 days.

- Windows unshaded, no night ventilation, and AC operation;
- Windows unshaded, full-power night ventilation (i.e., 3.2 ach), and AC operation;
- Windows unshaded, half-power night ventilation (i.e., 2.3 ach), and AC operation;
- Windows shaded, no night ventilation, and no AC operation.

Night ventilation was employed between 21:00 and 08:00 (local times), no sooner than when the outside air was sufficiently lower (2 °C to 3 °C) than that inside.

## 2.2. Data Analysis

Out of 13 temperature measurements received, only three will be assessed in the present work—namely, the outdoor, indoor, and fan (when needed) temperatures.

### 2.2.1. Uncertainty Analysis

The measurement uncertainties based on a 95% confidence level are listed in Table 1 on page 5. All errors are expressed as the percentage of their corresponding values. The root-sum-square method, as defined by Kline and McClintock [20], was used.

**Table 1.** Measured quantities, their typical values, and their uncertainties.

| Quantity | Typical Value | Measurement Error |
|---|---|---|
| Fan diameter, $D$ (mm) | 114 | $\pm1.75\%$ |
| Fan gap thickness, $h$ (mm) | 20 | $\pm5.00\%$ |
| Fan air speed, $u$ (m s$^{-1}$) | 6.7 | $\pm4.26\%$ |
| Temperature, $T$ (°C) | 30 | $\pm1.33\%$ |
| Energy consumption, $Q_{AC}$ (kW h) | 1.2 | $\pm2.94\%$ |

The surface of the fan, assuming a production tolerance, gave an uncertainty of 7.23%, and combined with the air speed's uncertainty, yielded an accuracy of 8.35% for the air changes per hour.

### 2.2.2. Raw Data Filtering

Since in this work, the data were not intended for a complex analysis (or for direct calculations), and will exclusively be called upon in an observatory kind of fashion, the removal of high-frequency components alone was deemed to suffice for the dataset's quality control.

Following this premise, to track temperature data more closely and to account for transient effects, the Savitzky-Golay filter (see Orfanidis [21]), also known as the smoothing polynomial filter, was used in the MATLAB environment [22] (pp. 329–344). The filter has a tendency to preserve the dominant frequency components of the data while smoothing it. Undesirable effects, such as noise due to sampling inaccuracies, will thus be avoided. An example of the filter's application can be seen in Figure 2 on page 5.

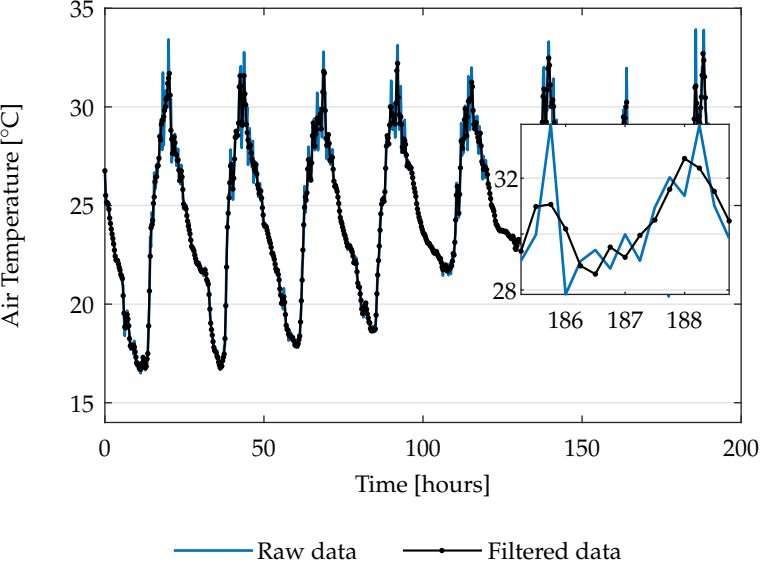

**Figure 2.** Raw and filtered temperature data.

## 3. Results and Discussion

As the main interest in this study was the effectiveness of night ventilation in providing for a low-cost way to cool structures, the main criterion chosen for evaluating the performance of the experiment was the decrease of cooling energy.

### 3.1. Mechanical Cooling Supplemented by Night Ventilation

In the first phase, the windows were unshaded. An air-conditioning unit was working throughout the period. Figure 3 on page 6 shows the daily indoor and outdoor temperature patterns of the building. During the first half, with no fan-temperature readings, only the air-conditioning unit operated as a cooling agent. Then, full-capacity night ventilation followed. The gap in the dataset, as indicated in the graph, was due to a power outage.

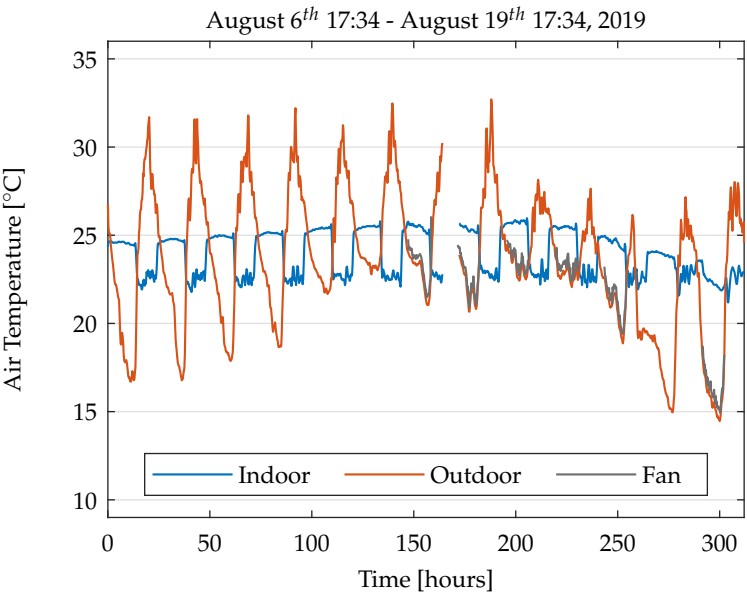

**Figure 3.** Daily outdoor and indoor temperature patterns of the building, with a working air-conditioning unit and unshaded windows, for two periods: first without, and then with full-capacity night ventilation.

This period was characterised by mild fluctuations of the outdoor average and diurnal temperatures, although near the end, rainy and windy days occurred. Seeing that the diurnal temperature was rather high for the non-ventilated part was rather unfortunate. Diurnal temperature, and high values of minima, constitute an efficient indicator for successfully corroborating night ventilation with mechanical cooling means, as previously mentioned.

After the first phase finished, the building was left to float freely for about 10 days, so as to alleviate any inertial effect superseding passive cooling. Consecutively, night ventilation was re-employed for a three-day period. The setup was the same, the only difference being that the night ventilation operated at half-capacity. As Figure 4 on page 7 shows, weather fluctuations scarcely occurred, but yet again, small diurnal temperatures and high minima values prevailed.

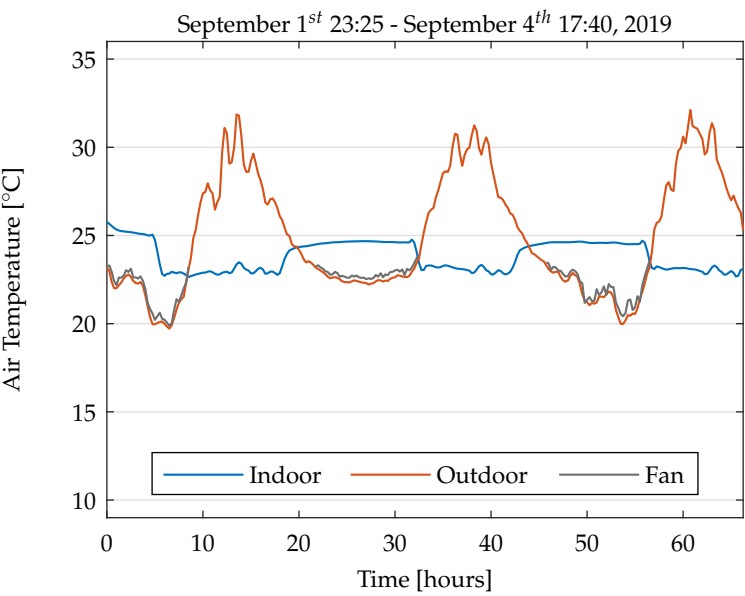

**Figure 4.** Daily outdoor and indoor temperature patterns of the building with a working air-conditioning unit, unshaded windows, and half-capacity night ventilation.

Figure 5 on page 7 shows the outdoor minimum, maximum, and average temperature for every day monitored, as well as the daily corresponding power consumption of the air-conditioning unit. A drop of the average cooling energy consumption, when night ventilation was at full capacity, below the non-night-ventilated period of about 27% per day was observed. At half-capacity, night ventilation yielded a drop of mechanical-energy-consumption below the non-night-ventilated period of about 15% per day.

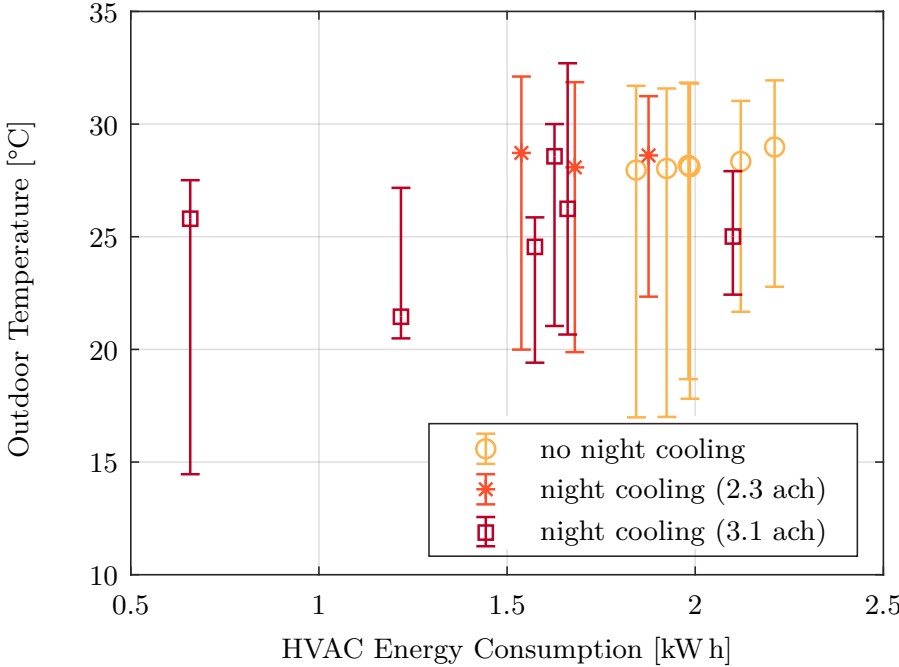

**Figure 5.** Cooling energy evaluation of night ventilation with respect to the daily average and diurnal temperatures.

However, to speak bluntly, the comparison of the average daily energy consumption between different days seems to be a fallacious one. Indeed, each single day is a very complex system,

exhibiting strong multi-dimensional behaviour, and it would make little sense to assume similarities. However, considering the outdoor temperature as the dominant comparison factor, in this case, one may assume that non-night-ventilated days required less cooling. Apart from the average outdoor temperatures being somewhat similar in both non-night-ventilated and night-ventilated days, the highest diurnal temperature values lay within the former. Thus, one would expect that the night-ventilated days would be in need of more cooling. The fact that the average cooling consumption was less testifies to the possible contribution of night ventilation in reducing cooling demand.

An interesting observation is that when night ventilation is employed in days with low diurnal temperature and minima above 20 °C, the energy consumption is similar to the values of non-night-ventilated days. As stated by other authors [4,5,16], night ventilation ought to be used with sufficient diurnal range and minima well below 20 °C, otherwise the structure acts as a heat trap, and may even reach higher minima values than that outside.

The two unexpectedly low energy values of the graph (i.e., 1.22 kW h and 0.66 kW h) succeeded a rainy night. Even during a rather hot summer, a rainy day can store enough cooling loads due to outer surface evaporation, and significantly reduce the cooling load demand of the proceeding day(s). In a similar fashion, a sunny day during cold winter helps with the required heating loads.

It would seem that night ventilation intensity is invariant of cooling demand reduction. Indeed, as previously stated, numerous studies [5,7,13] have shown that in order to achieve notable results, night ventilation ought to be employed at, 10 ach, at the very least. Nevertheless, the fact that cooling demand was somewhat decreased may be (and most likely is) attributed to the unusually high thermal mass of the structure, and its material's unique thermal properties.

### 3.2. Free-Floating Structure

In the second phase, the windows were provided with fixed, rolling, internal shades. The internal shades were black in color, made by a thermal insulated fabric with 100% blackout UV protection. No form of cooling was used. Figure 6 shows the daily indoor/outdoor, as well as the inner/outer surface temperature patterns of the shaded room.

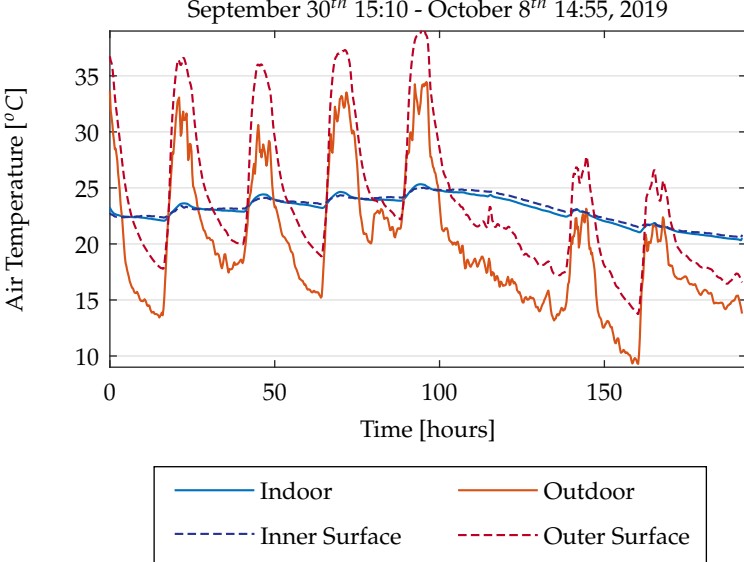

**Figure 6.** Daily indoor/outdoor and inner/outer surface temperature patterns of the building. The windows were shaded, and no cooling method (passive or mechanical) was employed.

The high-time lag (about 3 to 4 h) recorded testifies to the thermal stability that the building can provide. This observation is in agreement with the corresponding literature on earthen materials. Aimilios et al. [12] showed that a wall 50 cm thick yielded a time lag of approximately 5 h, while Baggs

and Mortensen [23] suggested that a 25 cm wall presented a time lag of 9.2 h. Both Asan [24] and Andreadaki [25] indicated a time lag of about 8 h for a 30 cm wall.

The beginning of the monitoring period was characterised by mild fluctuations of the outdoor average and diurnal temperatures, while the rest was defined by steep changes due to unsavoury weather conditions. Note that the indoor temperature, although of similar pattern to that outdoors, is strictly contained within the bounds of 20 °C to 25 °C.

A drop of the indoor maximum below the outdoor maximum of about 26% can be noticed at the 95 h mark. The indoor minimum temperature was also higher than the outdoor minimum by about 133% at the 160 h mark. The typical indoor temperature swing of the building was about 13% of the outdoor range. These data are in total agreement with Givoni's [1] observation of how a high-mass structure ought to behave while well-insulated and shaded.

Figure 7 on page 9 shows the cumulative density distribution of the same period. An increase of the indoor average mean temperature over the outdoor average mean of about 28% can readily be observed. This comes to show that adobe buildings can not only minimise heat stress during heat waves, but also maximise thermal comfort during cold waves.

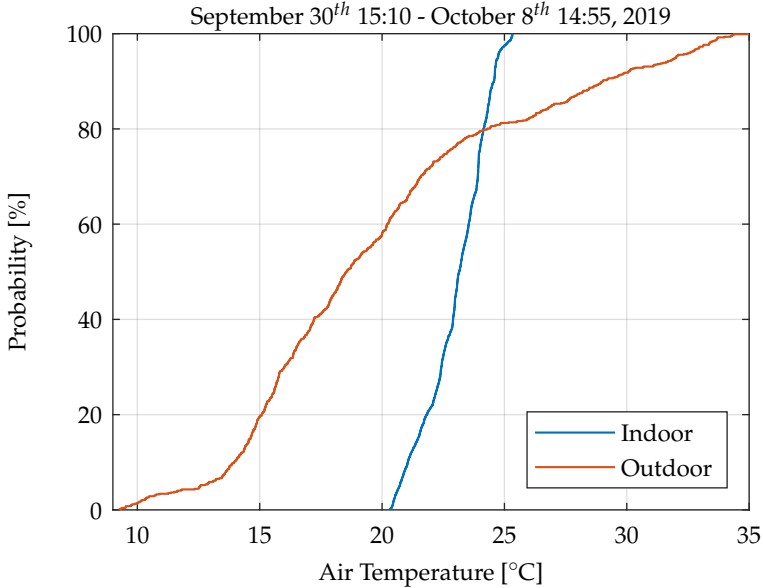

**Figure 7.** Cumulative frequency distribution of Figure 6 on page 8.

## 4. Conclusions

In this work, the collaboration of mass and night ventilation in decreasing cooling demand was experimentally investigated during the late summer period of 2019. In order to correctly estimate and control the airflow during night ventilation, an exhaust fan was used.

Both schemes where mechanical cooling was corroborated by night ventilation showed real promise. The average daily energy consumption decrease achieved varied from 15% to 27%. However, no clear correlation between night ventilation intensity and energy-saving could be established.

The building itself, when free-floating, maintained acceptable thermal comfort levels during unsavoury weather conditions. Its high thermal mass lowered the indoor maxima during a heat period and increased it for a cold period. Due to the inertial effect introduced by the structural material, the average indoor temperature swing was also 13% of the outdoor swing.

Night ventilation could be a good cooling strategy in regions where the diurnal temperature range is high enough to enable significant cooling storage, thereby lowering the need for mechanical cooling. Further, and more robust research is required to fully realise the building's potential for night ventilation.

Application of passive cooling systems alone will not bring the environmental cost of the world's buildings into line. However, "mixing mode" cooling systems may make the buildings' greenhouse gas policy more feasible, expanding the discourse from passive and mechanical cooling systems to an integrated variant of the two.

**Author Contributions:** Conceptualisation, M.Ç. and M.D.; methodology, M.Ç. and M.D.; software, M.D.; validation, M.D., M.Ç. and K.P.M.; formal analysis, M.D. and K.A.K.; investigation, M.D., M.Ç. and K.P.M.; resources, M.Ç. and M.D.; data curation, M.D.; writing—original draft preparation, M.D.; writing—review and editing, M.D., K.P.M., K.A.K. and K.-S.P.N.; visualization, M.D.; supervision, M.Ç.; project administration, M.Ç. and M.D. All authors have read and agreed to the published version of the manuscript.

**Funding:** This research received no external funding.

**Acknowledgments:** We are grateful to Ruhi Kafescioğlu for providing access to the adobe house, for otherwise the experiment would not have been possible. We are also extremely grateful to Duru Koçak, whose help with setting up the experimental equipment saved us innumerable hours of labour.

**Conflicts of Interest:** The authors declare no conflict of interest.

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
