# Peer review of "Utilisation of Mass and Night Ventilation in Decreasing Cooling Load Demand"

_sustainability, doi:10.3390/su12187826_

Round 1
Reviewer 1 Report
The case study and the experimental apparatus seem to be adequate to offer a further contribute to the night ventilation topic.
However, in my opinion, Authors should better rearrange their data to improve the readability on a daily base time and to clearly define the correlations among the night ventilation, the energy demand and also the internal comfort.
Author Response
Since no numerical extension of the results was possible at the time, and since the experimental periods for every case were rather short and uneven at times, the data presentation lacks rigour from a statistical point of view. Now the fact that night ventilation contributes to the decrease of mechanical energy is taken with a large grain of salt, that is the outdoor temperature swing is sufficient for a day-to-day comparison. Using this rather unsound approach makes it difficult to further analyse data, or use more advanced statistical models, since this assumptions will extrapolated and thus further complicate the results. So the way the data is presented seems to be the 'safest' option.
The term thermal comfort is used, when the structure is free-floating, in a broader context to construed the rather low indoor-temperature-swing. Thus no association is made with a theory to characterize the thermal comfort per say (such as Fanger's PMV). Desirable such an index may be, the problem is that the calculation itself requires, among other things, that the room's mean radiant temperatures be known, and hence all of the wall's surface temperatures. The present experiment, unfortunately, falls short of the necessary data to see through such an endeavour.
Reviewer 2 Report
The article deals with an experimental case study where the authors would like to verify the effectiveness of night ventilation in reducing the energy consumption of HVAC system for cooling a room of a high-mass building.
The abstract states that "night ventilation was undeniably impactful, reducing the average expected cooling-energy demand up to 27%", even if the graph in Figure 5 and its explanation shows that the strong reduction of the HVAC energy consumption is also due to the external climate conditions (rainy days), so the correlation between the use of nocturnal ventilation and HVAC energy consumption reduction is not clear.
Furthermore, the aim of the paper is not clear. The use of thermal mass for cooling energy reduction and night ventilation has already been widely studied in literature and it is also adopted in real constructions, so I do not understand what is the new concept and contribution added by this paper to scientific research. At the moment, it seems a simple and partial analysis of some experimental data, without a scientific literature framing (the introduction is really too short and it does not present the actual state of the art of the scientific literature on this topic) and so it is not suitable for publication.
Other comments:
- it would be interesting a plant of the building, not only a side view, to understand the positioning of the thermocouples.
- the characteristics of the walls (thermal transmittance of the walls, windows, time constant of the building, HVAC characteristics, position of the fan, types of shading of the windows) can help in understanding the behavior of the building
- it is not clear why 13 thermocouples were used but only 3 thermocouples data are analyzed. In particular, how the indoor temperature is chosen between the various thermocouples? Is the fan temperature important for the analysis, been similar to the external temperature, with a small difference (increase) due to the effect of the fan?
- I am not sure that three days of monitoring are sufficient to gather all the information of the dynamic behavior of the structure. The authors state that the building has a high mass, so probably a 7-day measurement (or more) for each type of experiment is needed
- it is not clear why the measurement errors are included in Table 1 but not used in the analysis.
- the comparison of the Free-Floating Structure with the Night-Ventilation data has little sense, as the two periods are different in terms of external temperature, so the structure has different thermal exchanges with the external environment. I think that also the solar irradiation incident on the building is different between the two cases. A comparison of the HVAC requirements, normalized on the mean external temperature and radiation, is needed to better understand the results.
- it is not clear how the authors present data of the late summer period of 2020 (August and September) while we are in July. I think that there is a typo.
Author Response
Indeed, since the correlation between night ventilation and energy consumption is not clear and is only commented upon an observatory fashion, a more down to earth language was used. line 9
Agreed, the vibrancy of literature concerning collaboration of night ventilation and high-mass buildings is overwhelming. However, not many works deal with the specific material (Alker). So, in that sense, the purpose of the paper is the collaboration of Alker with night cooling as a means of decreasing energy demand.
Floor plan of building added as a subfigure (b) of Figure 1. Paragraph added on the description of thermocouples' positioning and measurement method. lines 107 - 110.
Material's thermal properties mentioned between lines 87 - 90. Windows shadings' material mentioned at lines 200 - 201. HVAC power mentioned at line 100.
The rest of the measurements, that is the the two walls' temperature data, are part of an ongoing project to assess the material's (Alker) thermal behaviour and are thus beyond the scope of the present work. comment added at line 133
The way the data is presented, that is hours of operation, renders the use of the fan's temperature necessary as to indicate the operation intervals of night cooling.
Indeed, similar works have implemented night cooling for significantly larger time periods. However, unsavoury operating conditions (power outages, instrument malfunction, weather conditions etc.) made for the unevenly datasets.
Errors were provided for the sake of clearness of exposition. They were not intended (although they could have been) to be used in the data, but merely provide an estimation of the measurements' accuracy. The uncertainty analysis took place only for the determination for the air changes per hour, and a 8% uncertainty is a stringent enough result for the purpose of this study.
The free-floating structure was not intended to be compared with the night ventilation data. After all, they are placed in different subsections and their purpose (different in nature) is described in the introduction's paragraph. lines 66-71.
The irradiation incident might be a possibility, but no data for solar radiation was collected, thus there is no direct way of knowing.
Year typo fixed at line 219.
Reviewer 3 Report
Hi:
I feel the authors should provide the following:
- A plan schematic of the test house with the location of where the fan is placed
- No mention as to any windows being opened to allow ventilation at night, Need clarification.
- need to provide cross sectional schematics of the typical wall in this study along with the attic/ceiling cross section and mention the thermal properties of the different layers in these sections, i.e. thermal conductivity, specific heat, density of the different layers of the walls and ceiling/roof.
Author Response
Plan schematic added as a subfigure (b) of Figure 1 (between lines 92 and 93) with an indication of the fan's position.
For night ventilation the exclusive use of the fan was used while all windows were shut. Added the appropriate clarification at line 99.
Some of the building material's thermal properties were added, lines 88-90.
Reviewer 4 Report
In the current work is made an anlaysis of utilization of night ventilation. The topic is very interesting and important for our energy future. I have a suggestion to make more clear and visible figure 1b.
Author Response
Schematic of the structure (Figure 1b) has been somewhat improved; between lines 92 and 93
Round 2
Reviewer 2 Report
The new version of this article did not satisfy me. The main reason is that the authors only partially have answered to the many doubts that I have written in my previous review.
They added a floor plan and some information about the characteristics of the Alker material. However, there is no added information on the aim of the paper, comparison with similar papers using different materials but still discussing night ventilation and energy consumptions. The traditional "Discussion" section is, in that sense, completely absent. The authors state, in their response, that: "the vibrancy of literature concerning collaboration of night ventilation and high-mass buildings is overwhelming. However, not many works deal with the specific material (Alker). So, in that sense, the purpose of the paper is the collaboration of Alker with night cooling as a means of decreasing energy demand." and also "similar works have implemented night cooling for significantly larger time periods. However, unsavoury operating conditions (power outages, instrument malfunction, weather conditions etc.) made for the unevenly datasets." but there are no comparison of their work with other works found in literature.
Still, the use of a reduced three-day monitoring seems not sufficient for a structure with a high capacity.
In the end, I think that the paper is at the same level of the previous revision.
Author Response
Comparison with similar papers using different materials: In order to arrive at a rational, and robust, method for describing the efficiency of night ventilation, it is necessary to discuss the sources and nature of the parameters defining it. One useful clarification of these parameters is as follows: climatic, building, and technical. Thus, the answer to the comment “comparison with similar papers using different materials” cannot be given definitively since it is subject to the demand for the parameters to be the same for all cases (although further bibliography was added on the application of similar methods, lines 78-83). Hence it makes little sense to compare the present scheme to similar ones (using night cooling) since for either case the parameters are fixed (and differ in many ways). Even so, the scope of this paper was to somewhat experimentally assess the applicability of passive cooling techniques for the specific building, and not make a general statement about Alker per se. Apart for the brief description and the connection to carbon-free materials, Alker, was merely referred to within the context of a high-mass material.
No added information for the aim of the paper: The aim of the paper was made clear in the lines 66-71. It was experimentally shown that, for the specific building, the collaboration of night ventilation and mechanical cooling could potentially lead to a decrease of energy consumption for the latter. The comparison between the three different ventilation cases was somewhat validated on the premise of steady climate conditions (same average outdoor temperature etc.)
The use of a three-day monitoring period is not sufficient for building with high thermal capacity: The authors feel that the comparison of a three-day period, although not originally intended, should be sufficient for the purposes of this paper. A glimpse of the building’s behaviour may readily be seen at Figure 6, where the time lag between the outer and inner wall’s surface was 3-4 hours throughout a period defined by rather dynamic climate conditions (further references are added at lines 209-213). Also, considering the complexity that the climatic parameter bears, it is safe to assume that it will lessen the effect of “repeatability”. A study conducted by Aimilios et al. (12th reference), assessed an Adobe material’s thermal properties throughout a 24-hour period.
Round 3
Reviewer 2 Report
I still think that this paper does not add anything new to literature but a few data, even with a not-well detailed analysis.